# OpenReview forum: "Jailbroken: How Does LLM Safety Training Fail?"
_NeurIPS.cc/2023/Conference — NeurIPS 2023 oral_

### Official Review · Reviewer_VdLs · 2023-06-30

**Soundness:** 3 good
**Presentation:** 3 good
**Contribution:** 2 fair
**Rating:** 5
**Confidence:** 4

**Summary:**

This paper offers an insightful examination of adversarial misuse, or "jailbreak" attacks, against large language models (LLMs) such as OpenAI's GPT-4 and Anthropic’s Claude v1.3. By analyzing two proposed failure modes of safety training—competing objectives and mismatched generalization—the authors provide an empirical study of why these attacks can succeed and how they might be created.
Nevertheless, this paper could be critiqued for a lack of novel methodological or technical insights.



**Strengths:**

This paper provides a comprehensive examination of the vulnerabilities in large language models (LLMs) to "jailbreak" attacks.

**Weaknesses:**

I enjoyed reading the extensive experimental results and inspiring observations. The main reason I did not raise a higher score is that the paper seems to lack rigor in defining key concepts and quantifying its statements.

For example, the paper did not precisely define what a jailbreak attack is in the context of LLM. It is unclear what is safe/unsafe since that can be quite subjective. In making "analysis", e.g., "Competing Objectives" in Section 3.1, the paper only provided selected examples rather than formalism.

I feel the empirical studies are not sufficient to publish because there have been many existing empirical studies on jailbreaks and this paper does not distinguish itself from the rest with more in-depth formalism.

However, I could be biased. So I can adjust scores after reading other review comments and authors' responses.

**Questions:**

none

**Limitations:**

technical insights or novel method

---

> ### Author Rebuttal · Authors · 2023-08-09
>
> Thank you for your review of our paper, and for pointing out areas of concern. Here's our response to the issues you raised:
>
> 1. **Definition of jailbreak attacks.** We would like to highlight that we present a formal threat model for jailbreak attacks in Section 2.1 in terms of restricted behaviors. On the other hand, a completely mathematical definition of restricted behavior may be an unreasonable expectation, since judging harm (or other undesirable LLM output) is inherently subjective. Nonetheless, the potential of LLMs to cause harm is well documented [1, 2] and is a concrete concern for the deployment of such systems, especially as their capabilities improve in the coming years.
> To minimize the degree of subjectivity in our study, we defer the definition of harm to the model creators and follow a clear labeling scheme, detailed in Appendix B. In particular, we find that the tested models refuse to answer almost all of the harmful prompts without jailbreaking (see the performance of the "none" attack in Tables 1 and 3). This, along with the fact that the curated dataset was drawn from red-teaming efforts of the model creators, suggest that these prompts were considered harmful by the model creators.
>
> 2. **Related work on jailbreaks.** We are not aware of any empirical studies on jailbreaking language models of a similar nature as ours at the time of submission. If you know of any related work we may have missed, please let us know, and we will be sure to cite and discuss them.
>
> 3. **Conceptual contributions.** Finally, we would like to emphasize that our conceptual contribution of identifying the failure modes of existing methods goes beyond our empirical study. We believe these insights will push the field forward, as the weaknesses we identify cover almost all known attacks and highlight why current methods fail systematically. Indeed, as part of our responsible disclosure process we shared our results with affected model creators (OpenAI and Anthropic), and their feedback indicated that our results were not only novel, but also interesting and valuable to them.
>
> [1] https://arxiv.org/pdf/2112.04359.pdf
> [2] https://arxiv.org/pdf/2303.08774.pdf

---

> > ### Comment · Reviewer_VdLs · 2023-08-16
> >
> > The rebuttal has addressed most of my comments. I will increase the score.

---

### Official Review · Reviewer_qvjk · 2023-07-05

**Soundness:** 2 fair
**Presentation:** 3 good
**Contribution:** 3 good
**Rating:** 6
**Confidence:** 3

**Summary:**

The paper studies jailbreak attacks. More specifically authors hypothesize two failure modes of LLMs 1. competing objective and 2. mismatched generalization and base their discussion on why jailbreak attacks succeed based on these hypothesis made. They then quantitatively perform experiments on different jailbreak attacks and report the success of each.

**Strengths:**

1. The paper studies a timely and important topic.
2. The observations/discussion on competing objectives was interesting.

**Weaknesses:**

1. I would have liked to see more in depth and concrete discussions on what could be done to improve these systems against jailbreak attacks and even maybe the execution of this idea in small scale would have made the paper very strong.

2. The dataset discussion in section 2.2 could be more organized and clear with more detailed discussions to increase clarity.

3. I felt like the results do not directly support claims made in the paper in the sense that jailbreaks happen due to the two reasons 1.competing objectives And 2.mismatched generalization hypothesized by the authors. The quantitative results mostly show the success of the jailbreaks which is a nice finding on its own and mostly known, but they do not directly support the claims that these attacks are successful and happen due to competing objectives and mismatched generalization. Moreover, there might be other contributors not considered in this work. Although authors provide discussions and tie these jailbreaks to the hypothesized reasons in section 3, in my opinion analysis in section 4 does not strongly and directly support those claims. Maybe the paper needs to be motivated in some other way or written in a different manner so that claims are supported and reflected accurately in the experimental discussions and results.

**Minor comments:**
Typo line 288 ( such an attack is technically is beyond the scope of our threat model -> such an attack is technically beyond the scope of our threat model).


**Questions:**

Addressing my first and third concerns in the weaknesses section would be good.

**Limitations:**

Authors provided a discussion on limitations and broader impacts of their work which is appreciated.

---

> ### Author Rebuttal · Authors · 2023-08-09
>
> Thank you for the thoughtful and detailed review! Responding to each of the points you’ve raised:
>
>
> 1. **On defenses and small-scale experiments.** Towards better defenses, in Section 5 we argue that successful defenses may need to move beyond the existing pretrain-then-finetune paradigm. In this vein, the approach of Perez et al. to incorporate human preferences *during pretraining* [1] is a promising example that plausibly addresses both failure modes of standard RLHF.
> Furthermore, we highlighted white-box studies involving open-source models as a future direction in the Conclusion because no safety-trained open-source large language models existed at the time of writing, so there were no available small-scale models to experiment on. (The first such model, Llama 2 from Meta [2], was released after the end of the NeurIPS review period.) And even with open model weights, there still exists a significant gap between proprietary and open-source datasets/infrastructure for safety training. We nonetheless agree that further investigation with white-box access is an exciting future direction.
>
> 2. **On support for claims about competing objectives and mismatched generalization.** We would like to highlight that our ablations (in Section 4) of the simple example attacks (from Section 3) aim to pin down the mechanism of these attacks using only black-box access.
> For instance, for prefix injection, we show that the injected prefix matters for the success of the attack. Changing to an innocuous prefix reduces effectiveness significantly. This indicates that the autoregressively decoded prefix plays a key role in determining the refusal (or lack thereof) of the model. This autoregressive decoding behavior is a direct consequence of the pretraining objectives.
> And for Base64, we show that Base64 input suffices for a successful attack: just providing the input in Base64 suffices to escape safety training. This demonstrates that instruction following generalizes to Base64 inputs, but safety training does not. Given the success of RLHF, it is very plausible that including examples of Base64 inputs in the safety training set would lead to safety on this category of inputs. These together suggest that it is indeed mismatched generalization that drives the success of this attack.
>
> [1] https://arxiv.org/abs/2302.08582
> [2] https://arxiv.org/abs/2307.09288

---

> > ### Comment · Reviewer_qvjk · 2023-08-12
> >
> > I thank the authors for providing responses to my review. I am still not fully convinced about the discussions provided by the authors on the second point. If authors claim "Given the success of RLHF, it is very plausible that including examples of Base64 inputs in the safety training set would lead to safety on this category of inputs. " I would like to see some experiments on this and actual results to prove the point. I feel like there is lack of experimental and ablation studies to support some of the claims made.
> >
> > Regarding the comment 1 and it being deferred to future work, again this can add significant value to the paper and since currently it is missing from the current work, I am going to keep my score as is.

---

### Official Review · Reviewer_Q2oq · 2023-07-09

**Soundness:** 3 good
**Presentation:** 2 fair
**Contribution:** 3 good
**Rating:** 6
**Confidence:** 4

**Summary:**

This manuscript provides an initial exploration of the robustness of LLM systems against adversarial misuses, specifically focusing on "jailbreak" attacks. The authors successfully summarize existing threats, propose plausible hypotheses, and conduct empirical evaluations on three LLMs: GPT-4, Claude v1.3, and GPT-3.5. They identify two failure modes in the current safety training: Competing Objectives and Mismatched Generalization. The paper concludes by deriving valuable defense implications from their analysis and evaluations. Overall, this submission initiates a constructive discussion on the development of safe LLM systems.

**Strengths:**

The paper addresses a timely and significant topic. The authors carefully examined the problem, presented their hypotheses, and supported them with well-designed empirical experiments. Helpful defense implications are also provided following the analyses.

**Weaknesses:**

Certain sections of the writing could be refined to improve comprehension for a general audience. For example, the introduction to the current safety training mechanism and the general LLM training framework lacks sufficient detail, which may hinder readers' understanding of the concepts of Competing Objectives and Mismatched Generalization. Furthermore, the significance of the paper is constrained by the absence of direct access to detailed information regarding LLMs they use in their paper.

**Questions:**

1. Section 2.2 primarily focuses on the detailed description of the models and datasets used in the authors' evaluation. Will it be better placed in Section 4.

2. There appears to be some overlaps between Competing Objectives and Mismatched Generalization. For example, both the use of encoded output in Base64 and requesting unusual output formats may exploit prefix injection. Can you better explain the relationship between these two failure modes?

3. The description of the Adaptive Attack in Section 4.1 lacks clarity.


**Limitations:**

The author discussed the limitation arising from the lack of detailed information on LLMs, thus their hypotheses cannot be directly confirmed.

---

> ### Author Rebuttal · Authors · 2023-08-09
>
> Thank you for your detailed review and constructive feedback! To respond to the points you’ve raised:
> 1. **On background on LLMs and safety training.** Thanks for the suggestion—we will provide a more comprehensive review of background in the final version to make the paper more accessible for a broader audience.
> 2. **Competing objectives vs. mismatched generalization.** You are correct that these two categories are not mutually exclusive—successful attacks often combine strategies that exploit both. On the other hand, our ablation of the Base64 attack shows that encoded input alone suffices to break safety training in many instances. Thus, prefix injection is not a necessary component of the Base64 attack; the fact that the *input* has unusual formatting suffices.
> 3. **Clarification on adaptive attack.** We appreciate the feedback and will elaborate on the description of the adaptive attack in the final version: “To model an adaptive adversary who selects an attack based on the specific prompt, we implemented a straightforward `adaptive’ attack strategy. We consider this attack successful if any one of the 28 different evaluated attacks succeeds at eliciting an on-topic response to the harmful prompt.” Please let us know if this clarification addresses your question.

---

> ### Comment · Area_Chair_zbn4 · 2023-08-18
>
> Reviewer, please confirm that you read this rebuttal and adjusted your score and review if appropriate.

---

### Official Review · Reviewer_Nf9i · 2023-07-17

**Soundness:** 2 fair
**Presentation:** 3 good
**Contribution:** 2 fair
**Rating:** 6
**Confidence:** 4

**Summary:**

The paper evaluates GPT models and Claude models against Jailbreaks and comes up with two possible explanations for why jailbreaks are successful- 1. competing objectives between pretraining + IF finetuning and safety finetuning; and 2. pretraining+finetuning generalizing better than safety tuning. It also argues for safety-capability parity and that we need to move beyond the pretraining + post-training for safety paradigm to come up with more sophisticated methods for safety training.

**Strengths:**

Strengths:
1. Very clearly and well written; enjoyable read
2. Makes intuitive+compelling hypotheses around why jailbreaks are successful and makes them explicit and clear; provides evidence supporting the hypotheses
3. The safety-capability parity claim the paper makes is important and it's presented well with evidence for why its needed
4. I think the paper is significant given that it makes a compelling argument for why the current paradigm of safety finetuning won't scale and with good evidence for it. The evaluations and hypotheses are quite intuitive +  based on existing work but I think the presentation of this work in the paper and the narrative it creates is compelling and important.

**Weaknesses:**

Weaknesses:
1. Evaluations could be more robust and have a higher quantity of prompts
2. The main claims and methods are interesting and well-presented but not particularly surprising / novel
3. It should make clear that the two possible hypotheses are two reasons for jailbreaks but not the entire surface area of possible reasons


**Questions:**

It would be great if more data is added to evals and the evals are made transparent + available for everyone

**Limitations:**

Yes

---

> ### Author Rebuttal · Authors · 2023-08-09
>
> Thank you for your thoughtful review and feedback! We're glad you found the writing and hypotheses compelling. To respond to the points raised:
> 1. **On the number of prompts evaluated.** We acknowledge the desire for a more extensive set of prompts. However, our choice of 317 prompts for evaluation is consistent with other studies in this area, such as Shaikh et al. [1], who used 200 synthetic prompts. This number of prompts balances statistical power (see Table 2), API usage cost, and labeling effort.
> 2. **On data sharing.** We understand the importance of transparency and agree that wider access to our evaluation dataset could benefit the research community. Due to the potential for misuse (a concern shared by the ethics reviewers for this paper), we have been sharing our dataset upon request rather than openly releasing it. We are committed to ensuring access to any group with a legitimate research use and have already shared our dataset with several research groups in academia and industry who have reached out.
> 3. **On possible hypotheses for jailbreak.** You are right in that our mechanisms cover a large fraction, but not all, of the diverse set of known attacks. As in any security setting, there can be a long tail of possible vulnerabilities. We view our contribution as identifying prevalent pitfalls, similar in spirit to how OWASP maintains a list of the top vulnerabilities in web security [2]. We will clarify regarding this in the final version.
>
> Again, thank you for the review, and let us know if you have any further questions!
>
> [1] https://arxiv.org/abs/2212.08061
> [2] https://owasp.org/www-project-top-ten/

---

> > ### Comment · Area_Chair_zbn4 · 2023-08-18
> >
> > Reviewer, please confirm that you read this rebuttal and adjusted your score and review if appropriate.

---

### Official Review · Reviewer_XG6N · 2023-07-23

**Soundness:** 3 good
**Presentation:** 3 good
**Contribution:** 2 fair
**Rating:** 6
**Confidence:** 3

**Summary:**

This paper analyzes why safety training of LLMs fail and what could be the root causes for them. They offer two explanations: competing objectives and inadequate coverage of the model capabilities during safety training. They categorize existing attacks and propose some new ones into these two buckets, and evaluate GPT-4 and Claude v1.3. The results show that both the models are susceptible to jailbreaks.

Post-rebuttal comment: The authors addressed my concerns. I am happy to keep my score of weak accept.

**Strengths:**

I am broadly in agreement with the explanations put forth in this paper. The paper is written quite precisely and in a convincing manner. The experiments show that most of the 317 prompts constructed can be used in combination with one of the methods to successfully jailbreak.

**Weaknesses:**

The paper is mainly about evaluation of the models and the difficulty ahead of the defenses. While it is informative, it would have strengthened the paper to try some experiments and ablations on safety training (e.g., trying to remedy the mismatch in pretraining and safety-tuning in some ways). I can imagine the difficulty of experimenting with large, closed-source models, but some experimentation with small/medium open-source models would have helped.

My other concern is how much of the material in the paper is novel. Several of the existing attacks may have identified these or equivalent root causes and I am unable to clearly estimate how much of the paper's content is novel/surprising.

**Questions:**

What specific explanations and observations in the paper are novel when the existing attacks are taken in account?

Do you have any analysis of safety training interventions based on your identification of the key limitations of the current methods?

**Limitations:**

The authors have discussed the limitations and societal impact. They have also reported that they have disclosed their findings to the model authors and will coordinate the future release of the specific attacks with them to avoid harm. I appreciate these efforts.

---

> ### Author Rebuttal · Authors · 2023-08-09
>
> Thank you for your thoughtful review and feedback! To respond to the points you raise:
>
> 1. **On studying open-source models.** We highlighted white-box studies involving open-source models as a future direction in the Conclusion because no safety-trained open-source large language models existed at the time of writing, so it would not have been possible to study them. (The first such model, Llama 2 from Meta [1], was released after the end of the NeurIPS review period.) And even with open model weights, there still exists a significant gap between proprietary and open-source datasets/infrastructure for safety training. We nonetheless agree that further investigation with white-box access is an exciting future direction.
> 2. **On novelty / existing attacks.** While there are many known jailbreak attacks, our work stands apart in tracing the root causes of these attacks back to the design of training processes. This has let us uncover new attacks, guided by the principles we identify, with our best attacks outperforming those in informal public discourse. Unlike the informal literature, which typically involve a small number of handpicked examples, we provide a quantitative, systematic study of these attacks. Finally, we discussed our results with affected model creators as part of our responsible disclosure process, and their feedback indicated that our results were not only novel, but also interesting and valuable to them.
> 3. **On safety training interventions.** As discussed in Section 5, our findings suggest that robust solutions may have to come from beyond the existing pretrain-then-finetune paradigm. In this vein, the approach of Perez et al. to incorporate human preferences *during pretraining* [2] is a promising example that plausibly addresses both failure modes of standard RLHF. We view the development and evaluation of such approaches as an exciting domain for future research.
>
> Again, thank you for the review, and let us know if you have any further questions!
>
> [1] https://arxiv.org/abs/2307.09288
> [2] https://arxiv.org/abs/2302.08582

---

> > ### Comment · Reviewer_XG6N · 2023-08-14
> >
> > The response addresses my concerns, thank you!

---

> > > ### Comment · Area_Chair_zbn4 · 2023-08-18
> > >
> > > It appeared from your comment that you may have intended to raise your score--just checking that you did not forget.

---

### Official Review · Reviewer_PgD3 · 2023-07-25

**Soundness:** 4 excellent
**Presentation:** 4 excellent
**Contribution:** 3 good
**Rating:** 8
**Confidence:** 4

**Summary:**

The paper studies jailbreaking of large language models. The authors identify two categories of causes of jailbreaks, competing objectives and mismatched generalization, and use this insight to analyze existing jailbreaks and construct new ones. They empirically study the effectiveness of these jailbreaks on state-of-the-art models that have been trained to refuse unsafe instructions, and draw several conclusions from their analysis.

**Strengths:**

Jailbreaks pose an important safety problem for publicly-deployed AI systems. The conceptual analysis is novel and insightful, and the experiments are careful and thorough. The conclusion about safety-capability parity is especially interesting. Moreover, merely demonstrating extremely successful attacks directly helps companies deploying models to make their models safer via responsible disclosure. The paper is also very well-written.

**Weaknesses:**

The main weakness of the paper, which is easily rectified, is that the authors do not clarify (unless I somehow missed it) which if any models were queried as part of the construction of the attacks, or more broadly if the attacks have any particular intended target model. It seems reasonable to query a model when constructing an attack since the threat model allows this, but the information should be provided as it is needed to properly interpret the results of the paper. I don't think anything long and detailed is necessary if the process was somewhat ad-hoc, but some indication is necessary I think.

As a case in point, the paper interprets the results on GPT-3.5 (Table 3) as "GPT-3.5 not having the capability to understand complex inputs" (p. 8 line 336). However, I suspect that a more likely explanation of why the top 4 attacks are all attacks from jailbreakchat.com is that they specifically targeted at GPT-3.5 (since that version of ChatGPT is free-to-use and hence likely much more popular).

I also disagreed slightly with some of the claims in the section "What Scaling Won't Solve" (p. 8 line 350). The authors argue that scaling will not solve the problem of competing objectives, since the objectives will still compete. However, larger models may be able to obtain Pareto improvements on both objectives. Hence even though the competition between the objectives may remain, the models may still perform as well as necessary on the safety objective. Nevertheless, I did agree with the argument that scaling alone will not solve the problem of mismatched generalization (not counting using the models to generate stronger attacks, which is discussed later).

Minor point: in the phrase "targeted training is insufficient" (p. 8 line 325), the term "targeted training" is not defined. I initially interpreted it to mean "adversarial training" (which the results do not demonstrate is necessarily insufficient), whereas I think the authors mean "adversarial training that only targets a subset of the possible failure modes" (which it is unsurprising is insufficient, though I suppose it might still be worth noting).

**Questions:**

No questions other than those given above

**Limitations:**

No limitations other than those given above

---

> ### Author Rebuttal · Authors · 2023-08-09
>
> Thank you for the thoughtful review of our paper! Responding to the points you raise:
>
> 1. **On which models were queried.** Attacks were developed by querying GPT-4 and Claude on a subset of the 32 curated prompts, in line with the threat model described in Section 2.1. In addition, attacks from jailbreakchat.com were likely developed by querying GPT-3.5 and GPT-4, but not Claude, given the state of access at the time. The attacks for the 317 synthetic prompts were the same as those on the curated prompts, so they were not adaptive to the prompts (but they were implicitly adaptive to the model, as they transferred from testing on the curated prompts). Please let us know if this addresses your question–we will add this information to the paper in the final version.
>
> 2. **On GPT-3.5 attacks.** To clarify about GPT-3.5 on complex inputs, we observe that GPT-3.5 is unable to respond even to a harmless prompt for many of the more complex attacks—see Table 7 in Appendix D. (We will add this more specific pointer in the final version.) Nonetheless, we agree that the top jailbreakchat.com attacks were likely targeted so that GPT-3.5 could respond to them.
>
> 3. **On targeted training.** We used "targeted training" to refer to training aimed specifically at certain attacks. Specifically, it appears that Anthropic identified roleplay as a failure mode in their red teaming paper [1] and performed roleplay-specific training for Claude [2]. Our evaluation with Claude highlights that this approach did not address the underlying failure modes, and thus Claude remained vulnerable to other forms of attack.
>
> [1] https://arxiv.org/abs/2209.07858
> [2] See the documentation for “claude-v1.2” at https://web.archive.org/web/20230519130926/https://console.anthropic.com/docs/api/reference

---

> > ### Comment · Reviewer_PgD3 · 2023-08-12
> > **Response to rebuttal**
> >
> > Thank you for your rebuttal. This addresses my questions about these 3 points.

---

### Official Review · Reviewer_mfj9 · 2023-07-25

**Soundness:** 3 good
**Presentation:** 4 excellent
**Contribution:** 3 good
**Rating:** 5
**Confidence:** 4

**Summary:**

The paper investigates two reasons why jailbreak attacks against SOTA LLMs (GPT-4, GPT-3.5 Turbo, Claude v1.3) succeed despite extensive safety training: a) competing training objectives (safety objectives vs. pretraining/instruction tuning) and failure of safety training to generalize to conditions covered by pretraining/instruction training but not safety training. These two modes are then used as guiding principles to design new jailbreak attacks that are empirically evaluated against the above three models, starting from both known and newly synthesized harmful prompts; a large number of these attacks are shown to have a high success rate. The authors provide concluding hypotheses regarding future work to defend against attacks (more sophisticated safety models that match the models' basic capabilities).

**Strengths:**

The paper aims to examine jailbreak attacks in a more principled way and presents an empirical evaluation that is broader in scope than previous studies. Though the insights are not entirely surprising, the paper does a good job explaining and exemplifying the two failure modes, evaluating SOTA models along those axes, and providing quantitative results. Although the failure modes are intuitively clear and have been known informally by most developers of LLMs, it is valuable to see a concise formulation of the problems and a fairly thorough experimental study.The paper offers high-level suggestions for how to do red-teaming or safety training more effectively and as such would be valuable for the community.
The presentation is very clear; authors observe responsible disclosure practices and have discussed potential ethical considerations (recipes for creating jailbreak attacks).



**Weaknesses:**

1. The paper focuses on two failure modes only. Arguably these could be the most important ones, but the bigger question is whether one could come up with a more comprehensive taxonomy of failure modes. The paper doesn't discuss this in more detail.
2. Many of the observations and conclusions are by necessity tentative and unconfirmed, due to black-box only access to models. The authors could have verified their hypotheses more directly by using a smaller open-source model where more details about training conditions and data resources are available. This setup could also have been used to explore the space of failure modes more systematically (e.g., what role does decoding play?).
3. The most successful attacks were combinations, but those were not studied in detail from a conceptual point of view- this seems to merit an entirely separate discussion: since there's a combinatorial space of combined attacks, how could models be trained successfully to defend against these?
4. The 'safety-capability' parity point remains vague - what does this mean for the actual safety model, are there concrete mechanisms you could suggest to make models more 'sophisticated'?

It would be good to see more discussion of at least points #1 and #3 in the paper.

**Questions:**

It's not clear to me where the attacks from jailbreakchat.com (p. 7) land in Table 1 -- which ones does this map to?

**Limitations:**

The authors discuss the limitations (e.g., no access to models themselves, hence tentative conclusions) but in the light of these they could have opted to also include some targeted experiments with open-source models (relegated to future work in the paper).

---

> ### Author Rebuttal · Authors · 2023-08-09
>
> Thank you for the thoughtful and comprehensive review! To respond to the weaknesses and questions that you bring up:
>
> 1. **On the focus on two failure modes.** We view simplicity here as a strength rather than a weakness, as we identify two core issues that underlie almost all of the diverse set of known attacks. This is akin to how OWASP maintains a simplified list of the top vulnerabilities in web security [1]. We agree with you that a broader taxonomy could be envisioned—as in traditional security settings, there can certainly exist a long tail of exceptions. (E.g., OpenAI models can also be attacked via the system prompt / OpenAI's Chat Markup Language.)
>
> 2. **On black-box only access.** We highlighted white-box studies involving open-source models as a future direction in the Conclusion because no safety-trained open-source large language models existed at the time of writing, so it was not possible for us to investigate them. (The first such model, Llama 2 from Meta [2], was released after the end of the NeurIPS review period.) And even with open model weights, there still exists a significant gap between proprietary and open-source datasets/infrastructure for safety training. We agree that further investigation with white-box access is an exciting future direction.
>
> 3. **On combinations of attacks.** We believe that training should address underlying failure modes rather than specific attacks. Addressing these fundamental failure modes would likely render combinations of attacks ineffective. As an example, the approach of Perez et al. [3] to incorporate human preferences during pretraining could plausibly address both failure modes.
>
> 4. **On safety-capability parity.** By safety-capability parity, we mean that models performing safeguarding should be comparable (or stronger) in capabilities to the model being safeguarded. For instance, we envision language models as a necessary component of the safety pipelines of future systems, which contrasts with more primitive techniques like word filters.
>
> 5. **On jailbreakchat.com attacks.** The attacks from jailbreakchat.com are italicized in Table 1. Specifically, they are labeled as AIM, evil_system_prompt, dev_mode_v2, dev_mode_with_rant, and evil_confidant.
>
> We hope these clarifications address your concerns—we are also happy to engage in further discussions to enhance the paper. Thank you again for the insightful comments!
>
>
> [1] https://owasp.org/www-project-top-ten/
> [2] https://arxiv.org/abs/2307.09288
> [3] https://arxiv.org/abs/2302.08582

---

> > ### Comment · Reviewer_mfj9 · 2023-08-13
> >
> > Thank you for your rebuttal. It addresses my comments.

---

> > > ### Comment · Area_Chair_zbn4 · 2023-08-18
> > > **Did you mean to raise your score?**
> > >
> > > It appeared from your comment that you may have intended to raise your score--just checking that you did not forget.

---

### Decision · Program_Chairs · 2023-09-21

**Decision:**

Accept (oral)

**Comment:**

Reviewers all found this to be a sound and valuable contribution.